# Mitigating Catastrophic Risks and Food Security Threats: Effects of Land Ownership in Southern Punjab, Pakistan

**DOI:** 10.3390/ijerph17249258

**Published:** 2020-12-11

**Authors:** Awais Jabbar, Qun Wu, Jianchao Peng, Ali Sher, Asma Imran, Kunpeng Wang

**Affiliations:** 1College of Public Administration, Nanjing Agricultural University, Nanjing 210095, China; awaisnjau@gmail.com (A.J.); 2018209019@njau.edu.cn (K.W.); 2College of Economics and Management, Nanjing Agricultural University, Nanjing 210095, China; ali_2796@yahoo.com; 3College of Management Sciences, Comsats University Islamabad, Lahore 54000, Pakistan; drasmaimran@cuilahore.edu.pk

**Keywords:** risk management, food security, land ownership, multivariate probit model, Pakistan, Punjab

## Abstract

In the wake of environmental challenges, the adoption of risk management strategies is imperative to achieve sustainable agricultural production and food security among the Pakistani farmers of Punjab. For a deeper insight into farmers’ adaptive behavior towards climate change, this study explored the role of land tenancy in the adoption of risk management instruments, such as off-farm diversification, improved varieties, and crop insurance. Off-farm diversification was found to be a preferred instrument among landless tenants. The study also employed a multivariate probit model that further signified the role of land tenure in risk-related decisions. Apart from land tenancy, the results identified the prominence of risk perception, information access, and extension access in adoption decisions. This study also investigated the association between risk management approaches and food security indicators (household hunger scale, food consumption score). Analysis revealed a significant association between risk management tools and food security indicators.

## 1. Introduction

The modernization of agriculture is key to eliminating poverty and attaining economic goals, especially for agricultural economies like Pakistan [1,2]. The agriculture sector is the backbone of Pakistan’s economy, as it contributes nearly 19% and consumes 43% of the labor force [3]. The significance of agriculture in Pakistan’s economy is undeniable, but this sector is currently going through environmental degradation and catastrophic conditions [4]. Global ecological organizations have warned Pakistan’s government about the worst effects of environmental changes [5,6]. The country has seen back-to-back floods that caused significant devastation and uncountable losses to livestock, farming, and basic infrastructure [7]. According to different estimates, the economic impact has been more than USD 43 billion [8].

Additionally, untimely rains and heatwaves have endangered the food security situation of the farming community [9]. Farming is the most dominant source of income and is responsible for feeding the ever-growing population of Pakistan; it is imperative to secure it from environmental hazards and climatic uncertainty [5,6]. Previous studies have suggested the adoption of multiple risk management strategies to minimize climatic risks [10,11,12,13]. Risk management can be termed as an approach for agricultural risk reduction and enhancing food security [14]. The selection of these approaches is a challenging task as it involves a mix of environmental, financial, and marketing functions [15]. Hence, it is essential to implement a suitable risk management approach according to the situation. The proper implementation of risk management approaches improves the income and food security of farm households. Studies have identified crop insurance, climate-resilient varieties, and off-farm diversification as key strategies being practiced in Pakistan [8,12,15,16,17]. Crop insurance is regarded as one of the most widely practiced global risk management tools. Many developing countries face climatic disasters every year that cause significant loss to human lives and infrastructure facilities. Pakistan is facing the same issue, as the country faces devastating floods almost every year. Many researchers have ranked crop insurance as an efficient tool against floods and disasters [18,19,20,21]. The Pakistani government has launched a comprehensive program, the crop loan-insurance scheme (CLIS), which is obligatory for all farmers who seek agricultural loans. Multiple studies have highlighted the importance of ex ante risk-mitigating strategy, and crop insurance is a perfect example of this [16,22,23,24]. Moreover, diversification is an obvious approach to manage climatic shocks. Crop diversification and intercropping are used to minimize adverse weather effects and reduce pest attacks [25]. Apart from the farming aspects, households engage in multiple non-agricultural activities for the sake of income stability. Off-farm diversification was cited as an effective risk-mitigating strategy by various researchers [12]. Furthermore, improved and climate-resilient seeds can endure climatic shocks, leading to enhanced and sustained agricultural production [26]. These varieties negate harmful environmental effects such as water scarcity, heat stress, and soil nutritional deficiencies.

Land resource usage is considered to be the main determinant for livelihood, accommodation, and food security [27]. Land-tenancy agreements are mainly held responsible for influencing technology-related investment decisions that affect agricultural production. Land is regarded as a scarce resource, and its distribution and tenancy arrangements are often viewed as the crucial ingredient of any developmental strategy. Land tenure deals with the rights of an individual or community to land and other resources [28]. In other words, the land tenure system can be defined as a time-barred agreement that determines ownership (who can use the resource) and for how long (period), with other conditions attached to it [29]. Natural calamities disrupt human lives in multiple ways, and the impact of these calamities can be gauged through numerous aspects, such as the extent of property damage, individuals’ resilience to recover from damages, and land tenure security [30]. The farming community is one of the most vulnerable to natural disasters due to their sole reliance on agriculture for their livelihood and their insecure land rights. Farmers with weak tenure status are often forced to settle near a riverbank or other disaster-prone areas, and their land tenure rights are always subject to debate while making post-disaster decisions [31]. Hence, farmers with ownership status are less vulnerable than tenants are. Land tenure security is a crucial factor, as well as the allocation of aid and land restoration during post-disaster decisions. Thus, land tenure security improves the individual’s capacity to cope with natural disasters and catastrophes. It also promises incentives to invest in conservation practices such as agroforestry, irrigation, and soil-protection measures [32]. A common form of tenure agreement exists in most areas of Pakistan where the landowner also bears half of the cultivation costs with the tenant, sharing nearly half of the crop production, and is often called a sharecropper.

Moreover, another type of land tenancy agreement exists in the Pakistani context that is often called owner-cum-tenant, where farmers lease or rent another piece of land for further cultivation [13], whereas simple tenants do not own the land in any capacity and cultivate the rented or leased land. To the best of our knowledge, a very limited amount of studies have discussed the potential effects of land tenancy on climate risk management. Therefore, we investigated the impact of land rights on the selection of credit reserves, savings, and off-farm diversification in Khyber Pakhtunkhwa (KPK), Pakistan [13]. 

Apart from KPK, the southern parts of Punjab are considered some of the most disaster-prone areas of Pakistan. Furthermore, our study chose a unique set of risk management practices (crop insurance, improved varieties, off-farm diversification) that were not addressed in previous studies.

Multiple studies have discussed the effects of different risk management elements on food security [22,33,34,35,36,37]. Crop insurance directly and indirectly affects the food security situation of farm households. Firstly, crop insurance works as a shield against environmental hazards by the compensation against catastrophic losses, which directly improves food security. Secondly, it indirectly enhances farmers’ investment capacity in sustainable ventures [23]. We identified crop insurance as a key factor for the assurance of food security, as it mitigates the risk associated with agricultural activities. Hence, crop insurance provides protection from environmental hazards, which elevates production and ensures food security [38]. A study was conducted to investigate the association between improved maize varieties and food security. A generalized propensity score was used for analysis, and the results indicated a strong association between improved maize varieties and food security. Additionally, empirical evidence supports the relationship between livelihood diversification and food security [25]. Research suggests that the adoption of off-farm activities provides income stability and welfare to households. Moreover, diversification activities enhance a farmer’s capacity to absorb climatic shocks, which ultimately ensures food security [39]. Figure 1 illustrates the conceptual framework by depicting the flow of relations among internal and external factors, such as land tenancy, farmers’ risk perceptions, management, and their effect on household food security. No attention has been paid to the impact of risk management strategies on food security in Pakistan. Hence, to fill this research gap, our study aims to answer three research questions: (i) How does the land tenure system influence the adoption of risk management tools? (ii) What are the determinants of risk management strategies? (iii) How does the adoption of risk management tools affect household food security? 

## 2. Description of Variables

### 2.1. Dependent Variables

Crop insurance can be termed as a protection policy that covers potential losses due to unexpected natural events such as fires, floods, or droughts [10]. There are multiple government and private crop insurance programs which are available to Pakistani famers, such as the crop loan insurance option. The Pakistani government has launched a comprehensive loan insurance scheme for major crops such as cotton, rice, wheat, maize, and sugarcane. In contrast to traditional crop insurance schemes, the government supported CLIS works in a different manner. This scheme only applies to small farmer tenants or those who own land up to 25 acres. The farmers must obtain a loan from a government supported bank, and the government pays the premium. This scheme covers losses up to the loan amount, which is the largest drawback of this initiative. However, it lacks in many aspects compared to full-fledged crop insurance, but due to its prevalence and frequency in the selected region, we included it as crop insurance in our study [16]. Multiple studies have discussed the capability of crop insurance in alleviating climatic damage [23,24,40]. Crop insurance stabilizes farmers’ income by transferring risk to the third party and increasing the investment in agriculture. Hence, crop insurance indirectly affects a household’s food security status [22].

High-yielding varieties are usually characterized by attributes such as climate resilience, high yield, early maturity, and enhanced production quality [17]. In Pakistan, two institutes, namely, the International Maize and Wheat Improvement Center (CIMMYT) and the Pakistan Agricultural Research Council (PARC), are mainly responsible for introducing such varieties. Previous studies have highlighted the risk management abilities of stress-tolerant varieties. The adoption of such technologies mitigates climatic shock and enhances crop yields to ensure food security [34]. 

Off-farm diversification is a risk management strategy that deals with income fluctuations and mitigates risk at the farm level. Diversification is defined as a process where households invest in diversified activities to ensure their survival and improve their living standards. Farm households diversify their income for two reasons: to negate the adverse impact of climate on yield fluctuations, and to broaden their income sources to accumulate wealth for present and future investments [12]. Diversification can also be described as the poor’s reaction when agriculture fails to provide a sufficient livelihood [41]. The choice of off-farm activities depends on certain factors, such as age, education, social circles, and exposure.

### 2.2. Explanatory Variables

Our study used a multivariable probit model consisting of explanatory variables, such as household characteristics, farm characteristics, institutional variables, and risk perceptions, and their effect on the adoption of risk management strategies (off-farm diversification, crop insurance, improved varieties). The dataset of the study contains both categorical and continuous variables. Except for education, all variables were categorical. The selection of all these variables was based on empirical evidence. Household characteristics consisted of family size, education, and farming experience. The education variable was grouped into three categories, and each group was assigned a distinct value: primary = 1, eighth standard = 2, matric = 3. Primary, eighth standard, and matric are equal to five, eight, and ten years of education, respectively. Farming experience was the number of years during which an adult worked as a farmer. Moreover, family size was the number of adult members in a farming household. 

Institutional characteristics involved variables such as access to extension, information, urban linkage, and social participation. All of these variables were previously used in multiple studies related to risk perception and management [4,12,13,14,20,23,42,43]. Access extension was taken if the extension department visited or contacted in any form each month: if yes, then = 1; otherwise, 0. Information access was also taken as a dummy variable and consisted of the availability of information sources such as newspapers, television, and the Internet. Risk attitude was defined as the head of the household’s attitudes towards risk: if they were willing to take a risk = 1; if not, 0. Similarly, social participation was taken as a dummy variable that counted for the household participation in any community gatherings or cooperatives. 

Farm characteristics consisted of variables such as farm size, farm ownership, and owner-cum-tenants. Farm size counted as the present land size under cultivation. Multiple studies have indicated the crucial role of farm size in the uptake of risk management approaches [4,12,26]. Farm ownership deals with ownership status, which was taken as a dummy variable: if the farmer was the owner of the land = 1; otherwise, 0. Moreover, owner-cum-tenants were farmers who leased or rented another piece of land for further cultivation. Simple tenants do not own the land in any capacity and cultivate rented or leased land. A common form of tenure agreement exists in most areas of Pakistan where a tenant shares nearly half of the crop production, often called as a sharecropper. The landowner bears the other half of the cultivation cost. 

Risk perception is the awareness of climatic risks such as heavy rain, floods, pests and diseases, and droughts. A quantification technique was used to calculate risk perception, where respondents were asked to rank the severity of their risk perception. A Likert scale was used with a range from 1 to 5, where 1 represented the lowest perception and 5 represented the highest perception. Furthermore, farmer perception was transformed into a risk matrix where 6 represented a high risk, and 5 or below represented a low risk. Further, a dummy was created where 1 was used for increased risk, and for the 6 or above points, a 0 was used. The graphical depiction of risk perceptions is displayed below in Figure 2.

### 2.3. Food Security Indicators

Food security analysis was conducted using two indicators, the household hunger scale (HHS) and food consumption score (FCS). FCS was introduced by the World Food Program as a frequency weighted dietary-diversity score [15,44]. The household food consumption score provides all details of a household’s 7-day dietary pattern. Food consumption score is an indicator of 7-day food consumption and diet diversity. FCS calculation is based on a simple formula:(1)FCS=a1b1+a2b2+…a8b8,
where a = frequency (1 week recall period), 1–8 = food group, and b = weight. 

The weights of different necessary foods were as follows: sugar = 0.5; vegetables and fruits = 1; staples = 2; pulses = 3; meat, milk, and fish = 4. On the basis of multiple points, households can be categorized into the following groups: poor (<21.5), borderline (21.5–35), and acceptable (>35). Our second food security indicator, HHS, is often used to measure cross-cultural experiences and extreme food insecurity [15,45]. The HHS questionnaire contains three basic questions: (i) Was there ever no food in your household because there were inadequate resources to obtain more? (ii) Did you or any household member go to sleep hungry because there was not enough food? (iii) Did you or any household member go a whole day and night without eating anything because there was not enough food? Each response is valued as follows: 0 (twice a month), sometimes = 1 (3 to 10 times), and often = 2 (>10 times). Lastly, the scores were added up and categorized as follows: little-to-no hunger (scores 0–1), moderate hunger (scores 2–3), and severe hunger (scores 4–6). On the basis of empirical evidence [46], the association between risk management strategies and food security was found through a chi-squared test.

## 3. Data and Methods 

### 3.1. Site Description

This study was conducted in the Punjab province, which holds significant importance to the country’s economy because of its share in agriculture and exports. We selected two districts, Muzaffargarh and Dera Ghazi Khan (Figure 3), on the basis of vulnerability and food insecurity. Muzaffargarh is situated between the Chenab and Indus rivers, and Dera Ghazi Khan lies between the koh-e-Suleiman mountain range and the Indus river. The location of both districts has made them among the most disaster-prone spots in the entire province. Whenever these areas receive excessive rainfall, the overflow of water from the hills and the riverside causes flooding [47,48]. Apart from catastrophes, both districts lack basic healthcare and infrastructural facilities, and they are home to some of the poorest people in the whole province. Hence, in the absence of any adaptive mechanism, these districts are vulnerable to climatic extremes [7,16]. The Pakistan Disaster-Management Authority (PDMA) ranked these areas as Category A in terms of vulnerability, which shows their risk-prone nature [16]. Besides that, skewed land distribution favors large landlords, which is also a serious concern for the small farmers in these areas. Although agriculture is a mainstay for most people in these areas, a considerable number of people are associated with other industries. With that background in mind, the southern part of Punjab was selected for our study.

### 3.2. Data Collection

The rural areas of southern Punjab were our main focus for this study and the data were collected through 6 months of recent survey. On the basis of flood history and vulnerability, we selected two districts at the first stage. In the second stage, two tehsils (Tehsil: subunit of a district) were chosen from each district. In the third stage, we selected multiple villages on the basis of their disaster-prone nature and flooding history. As rural areas lack basic facilities such as the Internet and other infrastructure, we chose face-to-face interviews that were conducted by an experienced researcher instead of a web- or postal-based survey. A multistage random sampling technique was employed, and a primary dataset of 400 farmers was collected through this procedure. We used a 95% level of confidence and a 7% margin of error.
(2)$$n=\frac{N}{\left(1+Ne^{2}\right)}$$

### 3.3. Data Analysis

#### 3.3.1. Multivariate Probit Model

A multivariate model was applied to analyze the farmer’s adoption-related decision. The multivariate probit model (MVP) explains the interconnectedness among multiple risk management strategies better than univariate models. Relevant factors were chosen to check their impact on the adoption of risk management approaches. Farmers adopted a group of risk management strategies, and selection decisions were based on risk perceptions, risk attitudes, and socioeconomic factors. The MVP model can be explained as follows:(3)$$Y_{ij}=X_{ij}\beta_{j}+\varepsilon_{ij}$$

This equation represents a set of risk management strategies, where *X* is taken as a set of conditioning variables, $Y_{ij}$ is a latent variable related to individual *i* values and risk management strategies, and $\beta_{j}\;$is the vector of a parameter to be evaluated. The latent variables were assumed to have a linear combination with risk perceptions and socioeconomic variables ($X_{ij}$), which were expected to affect the simultaneous selection of risk management strategies alongside the disregarded characteristics. Stochastic error was taken as$\;\epsilon_{ij}$.
(4)$$Y_{1}=\alpha_{1}+X\beta_{1}+\varepsilon_{1}$$
(5)$$Y_{3}=\alpha_{3}+X\beta_{3}+\varepsilon_{3}$$
(6)$$Y_{2}=\alpha_{2}+X\beta_{2}+\varepsilon_{2}$$
where *Y1 **, *Y2 **, and *Y3* are latent variables for each risk management strategy; $Y_{ij}$ > 0 and $Y_{ij}\;$≥ 1, otherwise 0. The concurrent adoption of risk management strategies is obvious, so the probability for correlations among these decisions is expected. Hence, $\epsilon_{ij}$ elements face stochastic dependence, and ignoring this fact may lead to the biased estimation of probable choices. Moreover, in the multivariate probit model, multivariate normality and error terms mean that vector = 0 is assumed. With this assumption of multivariate normality distribution, the unknown parameters in Equation (2) were estimated through simulated maximum likelihood (SML) that used a Geweke–Hajivassiliou–Keane (GHK) simulator in evaluating multivariate normality distribution.

#### 3.3.2. Descriptive Profile of Respondents

The results of the descriptive statistics are presented in Table A1. The average family size of farmers was 6.53 persons per household. Most farmers primarily passed with an average score of 2.02, which shows the attainment of education among farmers. The results revealed that most farmers had substantial farming experience, which is reflected in the average farming experience of 11.31. Regarding farm characteristics, most farmers cultivated their land, as the average land ownership was 0.70. Nearly 26% of farmers had acquired extra land on lease or rent for further cultivation, and the average farm size was 5.4 acres. Almost 55% of the farmers had a risk-taking attitude. In contrast, 62% and 42% of the farmers had access to information and extension, respectively. Among risks, the perception of flood risks was a perceived risk with an average of 80%, whereas 56% of farmers were worried about heatwaves, and 41% were concerned about the increase in pests and plant diseases. 

## 4. Results

### 4.1. Role of Land Ownership in Risk Perceptions and Risk Management Strategies

The tables below represent the effect of land ownership on risk perceptions and risk management decisions. Figure 4 shows that the adoption of improved varieties is a major risk tool employed by farm owners to cope with risk, whereas land owners-cum-tenants also use this technique, followed extensively by off-farm diversification and crop insurance. 

Improved varieties seem to resolve the environmental stress issues of the farming community. Of the tenants, 44% were using off-farm diversification as the main risk mitigating tool. Tenants are usually perceived as more exposed to environmental stress than owners are. Hence, off-farm diversification seems an obvious choice to ensure the survival of farming households. Crop loan insurance and the uptake of improved varieties were adopted by 10% and 30% of tenants, respectively. Figure 5 shows that landowners were more sensitive regarding all risk perceptions. Of the farmers, 79%, 75%, and 70% were worried about floods, untimely rains, and pest attacks, respectively. Comparatively, farmers with owner-cum-tenant status were less sensitive about climatic risks, as 33%, 32%, and 30% of that category were concerned about floods, untimely rains, and pest attacks, respectively. Furthermore, farmers of all land ownership categories were risk-takers, with percentages of 68%, 72%, and 67%, respectively.

### 4.2. Correlation Coefficients of Risk Management Strategies

The results are displayed in Table 1. The estimation signs showed positive correlation between all three risk management strategies. The positive relationships among all strategies suggest that the adoption of one risk management approach influences the adoption of other risk management choices.

### 4.3. Determinants of Risk Management Strategies

The results of the interconnectedness discussed in Equations (2)–(5) are presented in Table 2. The findings indicate that the model fit, and the chi-squared test affirmed the significant variability explained by the explanatory variables in the adoption of all three risk management strategies. Hence, the findings confirmed our choice of using a multivariate model (Wald χ^2^ (39) = 104.97, *p* = 0.000). Concerning household characteristics, farming experience significantly predicted all risk management strategies. Besides that, farm ownership also predicted the adoption of all risk management strategies. 

Furthermore, during analysis, risk perception emerged as the strongest predictor of risk management strategies, as all of the risk perceptions were positively and significantly associated with the adoption of multiple risk management approaches. The perception of floods was positively correlated with the adoption of all available choices, as with a 1 unit increase in flood risk the chances to adopt off-farm diversification, improved varieties, and crop insurance increased by 0.43, 0.33, and 0.60 units, respectively. Similarly, the perception of heatwaves also predicted the adoption of all three risk management strategies. Moreover, the perception of untimely rain positively predicted the adoption of improved varieties and crop insurance. The perception of pests and diseases significantly influenced the adoption choice of improved varieties. Furthermore, access to information and extension were both significantly and positively associated with the adoption of all three risk management approaches. This shows that a 1 unit increase in access to information would lead to the adoption of off-farm diversification, improved varieties, and crop insurance by 0.30, 0.58, and 0.40 units. Similarly, a 1 unit increase in extension access would lead to the adoption of off-farm diversification, improved varieties, and crop insurance by 0.47, 0.66, and 0.56 units.

### 4.4. Risk Management Strategies and Food Security

Table 3 shows the chi-squared results, which indicate a significant association between the adoptions of off-farm diversification, crop insurance, and improved varieties, and both food security indicators. Regarding improved variety adoption, 57% of the farmers that fit in an acceptable category adopted improved varieties, compared to 51% in the borderline and 41% in the poor class. 

Furthermore, with regard to the HHS indicator, 71% of the farmers who had adopted improved varieties were in the little-to-no-hunger category, in contrast to 45% in the moderate hunger category and 29% in the severe hunger category. The findings in Figure 6. Also revealed that, regarding off-farm diversification, 57% of the adopters were in the little-to-no-hunger category, against 43% of adopters in the moderate hunger category and 29% in the poor class. With regard to the FCS indicator, 39% of the adopters were in the acceptable category, in contrast to 41% in the borderline and 29% in the poor category, as shown in Figure 7. Moreover, regarding crop insurance, farmers with 20% insurance were in the little-to-no-hunger category, whereas in the FCS category, 36% of crop insurance adopters were in the acceptable category.

## 5. Discussion

This research was conducted in the southern parts of Punjab, Pakistan to explore the determinants of risk management strategies and whether the adoption of these risk-mitigating approaches affects the food security situation of households. Multivariate analysis was applied to identify the main determinants of risk management strategies; further, chi-squared analysis was conducted to examine the association between adopted risk management tools and food security indicators. Moreover, the adoption intensity of risk management instruments and risk perception concerning land ownership were explored during analysis.

### 5.1. Effects of Land Tenancy on Risk Management Decisions

The results signify the crucial role of land ownership in adopting three risk management instruments to alleviate the negative effects of climate risks. Tenants were more likely to adopt off-farm diversification as a risk management strategy in comparison to farm owners. Hence, the lower adoption of off-farm diversification among landowners seems related to tenure security. Likewise, [13] reported a similar relation between off-farm diversification and land tenancy. Land ownership is associated with better household welfare and increased control over land-related decisions. Furthermore, landowners were more likely to adopt crop insurance than tenants were. Because of the disaster-prone nature of the study area, most of the farmers were more likely to adopt the crop loan insurance plans than other insurance options. As discussed earlier, the government launched a crop loan insurance scheme for disaster-prone areas where farmer tenant owners with up to 25 acres of land can access the insurance [49,50]. Although other insurance packages are also available for the farmers, tenants seem reluctant to adopt this risk management tool due to its poor adaptive capacity. Moreover, landowners were more likely to adopt stress-tolerant varieties in contrast to tenants. Having a strong backing of fixed assets provides financial wellbeing to landowners, which helps them to access better institutional coverage. Hence, because of the knowledge and capital-intensive nature of improved varieties, landowners are more likely to adopt this risk management strategy.

### 5.2. Factors Related to Crop Insurance Adoption

The results showed a significant association between crop insurance and farming experience, which indicates that greater farming experience increases the adoption of crop insurance. These findings are consistent with those of Ashfaq et al. (2008) [42], who also suggested the importance of farming experience, the realization of climate uncertainties, and choosing the right tool for managing catastrophes. Our findings are also in line with those of Mesfin et al. (2011) [51].

The findings revealed a significant relationship between land ownership and crop insurance adoption decisions, which confirmed that land ownership status affects risk management-related decisions. Hence, the land owners were likely to adopt crop insurance. These results are consistent with those of multiple studies [18,52]. Similarly, Fallah et al. [53] suggested the positive relationship between these two variables. Our findings are in contrast to those that found a positive relationship between land rights and a willingness to purchase decisions, whereas the authors of [21,54,55] found an insignificant relation between these two variables.

Access to information is positively and significantly related to crop insurance adoption as a risk management tool, which shows the awareness level of the farmers. Government agencies play a large role in the promotion of such schemes. Awareness level plays a crucial role in determining the right choice for risk mitigation. Our findings are in line with those of multiple studies [19,55,56].

Risk perceptions play a key role toward risk management decisions, as they can be termed as the main condition for managing risks [14]. Our results showed the positive and significant relationship between the risk perceptions of heatwaves, pests and diseases, and floods. Agriculture is the main source of income in rural households, and all these risks directly impact the agricultural productivity and food security of individual households. Multiple studies have reported the significant relationship between risk perceptions and adoption decisions. This is similar to Ullah et al. (2015) [4], who suggested that risk perceptions and risk attitudes are crucial factors for investing in risk management strategies. 

Moreover, extension access significantly and positively predicted the adoption of crop insurance as a risk management tool. This is similar to previous studies that also reported the positive relation between extension access and crop insurance [40,43,57]. The authors in [23] also suggested that extension contact disseminates the importance of crop insurance, which encourages farmers to adopt it. 

### 5.3. Factors Related to Adoption of Improved Varieties

Farmer experience is significantly associated with the adoption of improved varieties as a risk management tool. With increasing age and experience, farmers learn new techniques that suit them and facilitate agricultural production. Similarly, Islam et al. (2012) [58] found significant linkage between experienced farming and the adoption of improved varieties. 

Further, access to information was significantly and positively associated with the adoption of improved varieties. Having access to information channels educates farmers about the availability of the latest improved varieties with extreme climate-resistant features. Similarly, Kaliba et al. (2018) [43] suggested the improvement of communication channels to improve the uptake of improved varieties. Simtowe et al. (2019) [59] also reported the positive linkage between access to information and the adoption of improved varieties. 

Furthermore, agricultural extension access significantly and positively predicted the adoption of improved varieties. Agricultural extension visits disseminate information about the availability of the latest and improved seeds in the market, which encourages farmers to adopt certain varieties. Similarly, [60] reported positive linkage between extension contact and the uptake of improved maize varieties in Tanzania. 

All risk perceptions were significantly associated with the adoption of improved varieties as a risk management instrument, which shows that farmers seek to resolve their climatic issues through the adoption of stress-tolerant varieties. Jabbar et al. (2020) also found a significant relationship between climatic variation and the adoption of improved seeds [17]. 

### 5.4. Factors Related to Adoption of Off-Farm Diversification

Farm size significantly and positively predicted the adoption of farm diversification as a risk management tool. Small fragmented farms are often associated with resource mobilization and low-productivity issues. Hence, positive correlation seems logical and is supported by previous studies on this issue [61,62,63]. Results suggest that a decrease in farm size might push farmers towards the adoption of other sources of income. 

The results show the importance of land rights in choosing off-farm diversification as a risk-coping tool. Farmers with secure land rights invest in more diversified ventures such as livestock business or fruit cropping. Further, land security encourages farmers to rent out land that they are not using. Hence, in both cases, farmers with secure land rights are more likely to adopt off-farm diversification [64]. Similarly, Ullah et al. 2019 [12] found the same relationship.

Access to information plays a vital role in disseminating information about diversification opportunities and risk-coping tools. Likewise, our results support this notion and indicate the positive association between access to information and off-farm diversification decisions [65]. Asfaw et al. (2017) also support this directionality of association.

Off-farm diversification was significantly and positively related with all purposed risk perceptions. Multiple catastrophes continuously damage farmers’ livelihoods in different ways that leave no other option for farmers but to diversify their sources of income. Researchers suggest [12,13] the positive relationships among risk perceptions and off-farm diversification decisions. 

### 5.5. Association among Risk Management Strategies and Food Security Indicators

The results of chi-squared analysis revealed a significant association between such strategies and both food security indicators. The findings were consistent with those of previous studies, where a significant association was found between improved seeds and food security. In another study, Kassie et al. (2011) [66] revealed an increase in income and yield with the adoption of improved groundnut varieties. Moreover, Kijima et al. (2008) [67] suggested the valuable role of stress-tolerant varieties in reducing poverty. In [68], the authors found that communities that applied the latest technologies in Madagascar’s agriculture enjoyed high yields, poverty alleviation, and better food security. Asfaw et al. (2012) [69] also found a correlation between the adoption of climate-resistant varieties and the household welfare of Tanzanian farmers. Multiple studies have highlighted the role of insurance in securing food security [23,24]. Researchers have suggested that crop insurance provides stability to farmers and acts as an incentive to adopt the latest technologies. Agricultural insurance absorbs risks related to agricultural production, and facilitates farmers borrowing more finances and investing in other activities. A positive linkage between agricultural insurance and food security has also been suggested [24]. A positive association between rice yield and crop insurance has been found [23]. Moreover, livelihood diversification is associated with food security status. Bayero et al. (2019) found that livelihood diversification helps farmers to manage the required finances for routine operations [70]. Further, it was suggested that governmental involvement increased livelihood options, which elevated food security levels in Hurungwe district, Zimbabwe [41]. Mensah (2014) also indicated the importance of livelihood diversification in resolving food security issues [39]. 

## 6. Conclusions and Policy Implication

This study was designed to investigate the effects of land tenure on the adoption of risk management strategies, and to assess the association between risk management strategies and food security indicators. The results revealed that a substantial difference exists between landowners and tenants regarding risk perceptions and the adoption of risk management strategies. Analysis highlighted that off-farm diversification is a preferable approach for tenants, whereas landowners preferred the two other risk management instruments. Both of these instruments are capital- and knowledge-intensive in nature, which shows the relevance of adaptive capacity in adoption decisions. The results also stressed the importance of institutional variables such as extension and information channels on risk management decisions. Both of these factors disseminate crucial information about risks and their solutions. Although rural areas lack in infrastructural facilities, governments and stakeholders still need to find more accessible communication channels to boost adaptation levels against climatic uncertainties. The findings further indicate the significant association between risk management strategies (crop insurance, improved varieties, off-farm diversification) and both food security indicators (HHS and FCS). Hence, comprehensive policy (Figure 8) is required to elevate the adoption of these risk management approaches. The governmental CLIS scheme only covers and secures losses up to the loan amount. Henceforth, the government should launch a full-fledged crop insurance program that can compensate farmers up to the actual damage amounts.

Moreover, a policy is required to improve land-right security, which would ensure the overall food security of farm households. The adoption of risk management strategies works as a shield against climatic and market-based fluctuations, which helps farm households to achieve investment efficiency and improve overall welfare, including food security. This study lacks geographical coverage, as it only covered the southern part of Punjab. Punjab is a large region that contains multiple agro-ecological zones, and future research may compare the role of land tenancy in climatic risk adaptation across different ecological zones. 

## Figures and Tables

**Figure 1 ijerph-17-09258-f001:**
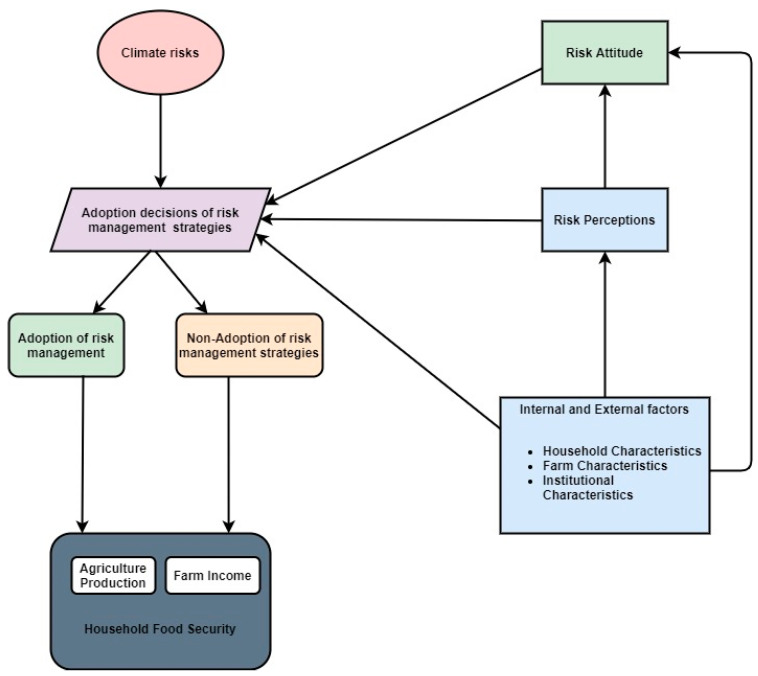
Conceptual model.

**Figure 2 ijerph-17-09258-f002:**
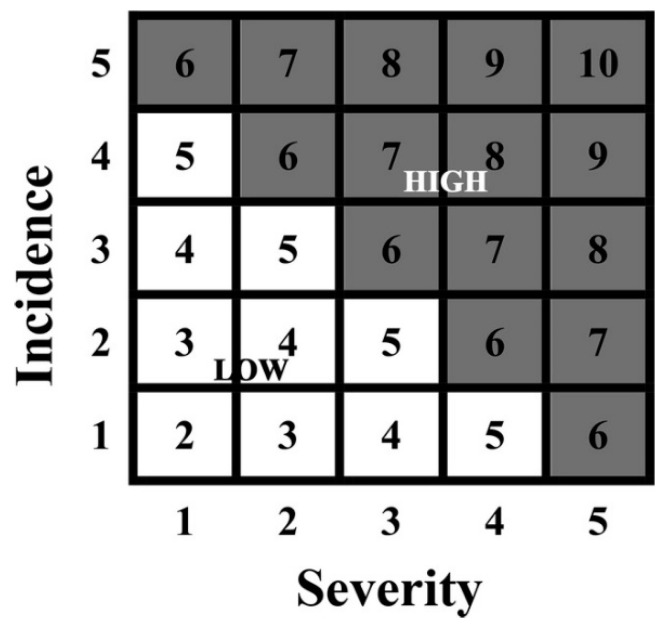
Risk perception matrix.

**Figure 3 ijerph-17-09258-f003:**
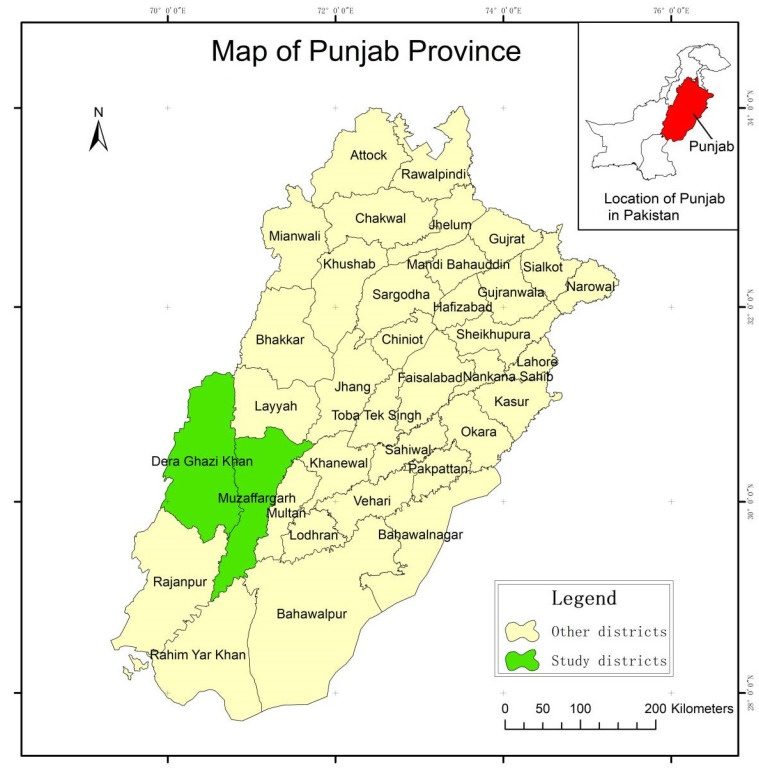
Study-area map.

**Figure 4 ijerph-17-09258-f004:**
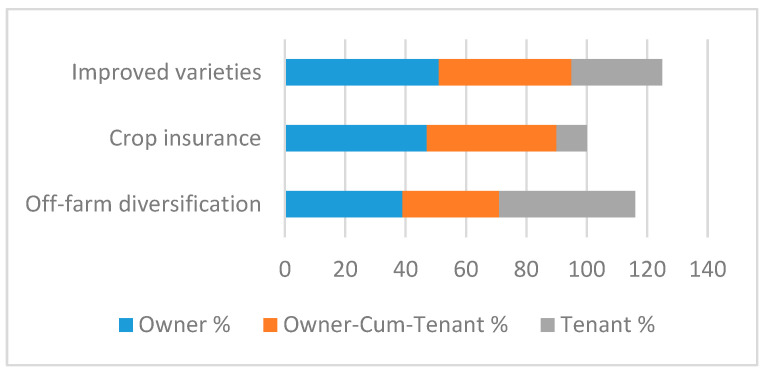
Risk management by land tenancy.

**Figure 5 ijerph-17-09258-f005:**
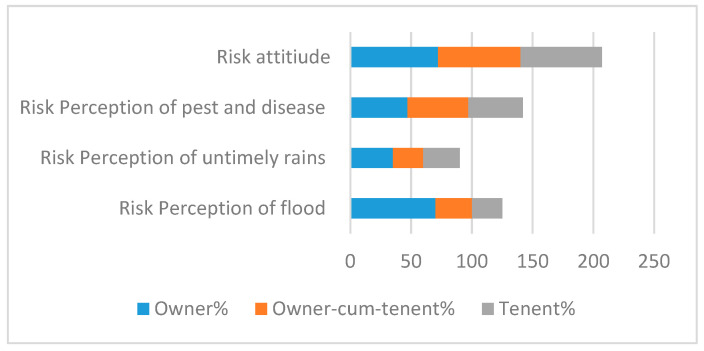
Risk perception and attitude by land tenancy.

**Figure 6 ijerph-17-09258-f006:**
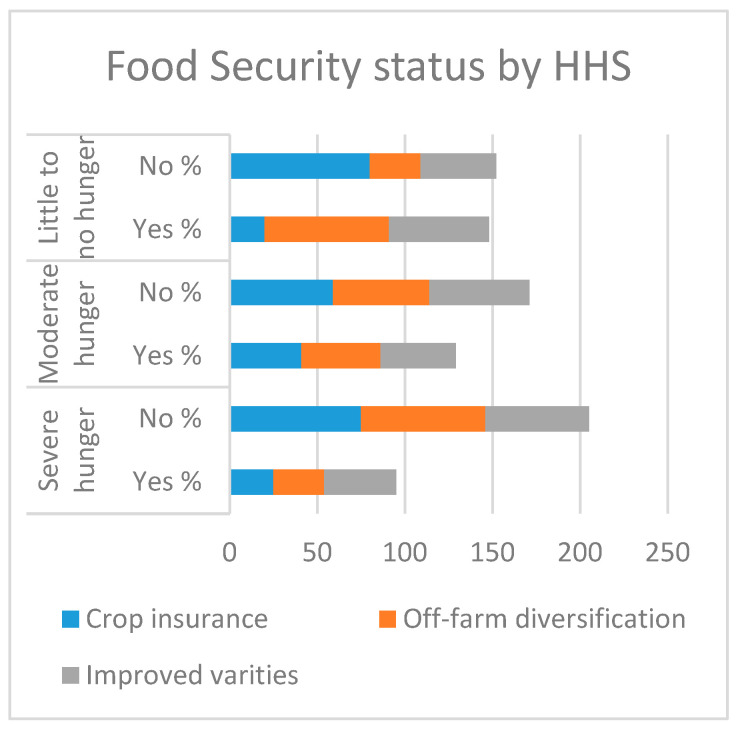
Household hunger status.

**Figure 7 ijerph-17-09258-f007:**
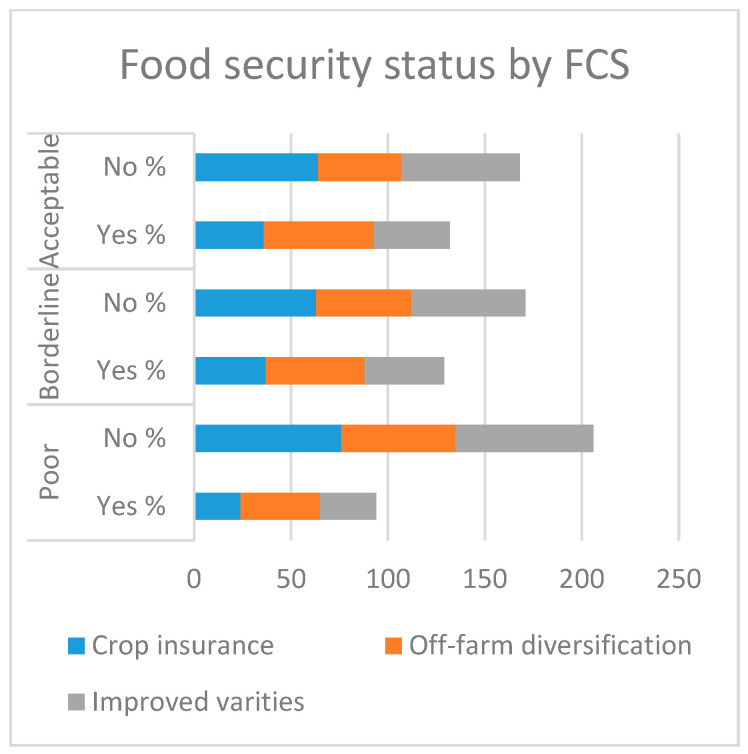
Food consumption status.

**Figure 8 ijerph-17-09258-f008:**
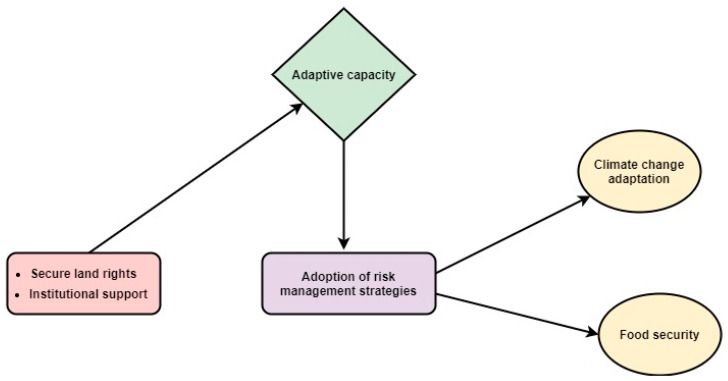
Policy implementation framework.

**Table 1 ijerph-17-09258-t001:** Correlation coefficients of risk management instruments.

Risk Management Combination	Coefficients
Off-farm diversification and crop insurance	0.65 ***
Off-farm diversification and improved varieties	0.84 ***
Improved varieties and crop insurance	0.77 ***

*** significance level at 1%.

**Table 2 ijerph-17-09258-t002:** Multivariate probit (MVP) model for estimating determinants of adoption of risk management tools.

	Off-Farm Diversification	Improved Varieties	Crop Insurance
**Household Characteristics**			
Family size	0.020	0.039	0.036
	(0.036)	(0.038)	(0.026)
Education	−0.031	0.109	0.110
	(0.107)	(−0.163)	(0.108)
Farming experience	0.007 *	0.010 **	0.007 *
	(0.004)	(0.004)	(0.003)
Risk willingness	−0.159	0.164	0.163
	(0.168)	(−0.104)	(−0.145)
**Farm Characteristics**			
Farm owner	0.386 **	0.430 **	0.323 **
	(0.153)	(0.159)	(0.158)
Owner-cum-tenants	0.153	0.162	0.156
	(−0.133)	(−0.150)	(0.027)
Farm size	0.026	0.006	0.018
	(0.029)	(0.030)	(−0.015)
**Institutional Characteristics**			
Access to information	0.300 **	0.584 ***	0.405 **
	(0.146)	(0.147)	(0.150)
Social participation	0.132	0.136	0.138
	(0.141)	(0.153)	(0.173)
Extension access	0.477 **	0.666 ***	0.560 ***
	(0.134)	(0.139)	(0.138)
**Risk Perceptions**			
Perceptions of floods	0.431 **	0.334 **	0.605 ***
	(0.173)	(0.182)	(0.186)
Perceptions of untimely rains	0.377 **	0.395 **	0.165
	(0.167)	(0.165)	(0.216)
Perceptions of heatwaves	0.327 **	0.339 **	0.137 ***
	(0.135)	(0.135)	(0.016)
Perceptions of pests and disease	0.133	0.293 *	0.236
	(0.188)	(0.136)	(0.224)
Constant	0.428	0.443	0.428
	(−1.887)	(−2.60)	(−2.47)
Number of observations	400		
Log pseudo-likelihood	−432.250		
Wald chi-squared	471.396 ***		

Note: standard errors presented in parentheses; ***, **, and * indicate significance at *p* ≤ 0.01, *p* ≤ 0.05, and *p* ≤ 0.1, respectively.

**Table 3 ijerph-17-09258-t003:** Chi-squared results of food security. FCS, food-consumption score; HHS, household hunger score.

	FCS		HHS	
	χ^2^	P	χ^2^	P
Crop insurance	21.49	0.000	7.35	0.025
Improved varieties	13.68	0.008	5.41	0.067
Off-farm diversification	17.62	0.001	6.15	0.046

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
