# Peer review of "Mitigating Catastrophic Risks and Food Security Threats: Effects of Land Ownership in Southern Punjab, Pakistan"

_ijerph, 2020, doi:10.3390/ijerph17249258_

Round 1

Reviewer 1 Report

The paper seems really ambitious and tries cover a broad area of research questions around how farmers adopt risk management instruments and associated drivers in Pakistan. The paper provides a nice introduction of the issues to explore. Though the paper is well structured, I do not think the core ideas are well communicated. And, thus, the contributions are not sharply articulated.

Please provide more explanation for the crop loan insurance scheme. Does it work similar to a regular crop insurance?

In many places, you need to still add author (year) after you used numbered in-text reference. And please proofread the paper to make sure small typos and improper capitalization are corrected. E.g., lines 103, 181, 181, etc.

You need to reference and explain equations provided in text.

In Equation 6, consider using summations.

The discussions around food security indicators are not clear. By food security, do you mean the welfare of farmers? It needs a lot more clarification since food security may have many different meanings in the literature.

Please revise all the figures following publishing requirement.

In Fig. 1, are the share supposed to sum up?

In general, the results are not well explained in a way supporting your discussions. For example, in Table 2, the results for Farm owner are significant. Does it mean farm owners tend to use risk management tools more? You need to explain the coefficients more quantitively rather than just describe them in discussions.

You mentioned in many places the results are in line with previous study. Then, what is the core contribution of this study?

You mentioned sharecropper a couple times in the paper. Was it in your analysis as well?

Overall, I think there are valuable points being raised in this study, with the utilization of survey data in two small regions in Pakistan. However, it would be very helpful if the authors can outline several research questions in the beginning and sharp the discussions of the results around the questions. Particularly, given the title, I was expecting more discussions on the impacts of land ownership in climate impact mitigation or adaptation. Because the land ownership (along with improved varieties) seems related to broader questions of how change the use of land would play key role in adaptation, which would not be considered in traditional production-function analysis, e.g., discussed in Mendelsohn et al. (1994) and more recently Zhao et al. (2020). Other related factors are adequately covered in this study might be the roles of (climate impacts on) land productivity and land conversion/transition cost, which might also affect risk managements.

Mendelsohn, Robert, William D. Nordhaus, and Daigee Shaw. "The impact of global warming on agriculture: a Ricardian analysis." The American economic review (1994): 753-771.

Zhao, Xin, Katherine V. Calvin, and Marshall A. Wise. "The critical role of conversion cost and comparative advantage in modeling agricultural land use change." Climate Change Economics 11.01 (2020): 2050004.  

Author Response

R

Response to Reviewer 1 Comments

Response to the general question: Respected reviewer thank you so much for your precious time and providing us your strong feedback to make our paper more productive. We tried to make our discussion more creative and have also added a new paragraph to highlight the other important aspect of the results.

Point 1: In many places, you need to still add author (year) after you used numbered in-text reference. And please proofread the paper to make sure small typos and improper capitalization are corrected. E.g., lines 103, 181, 181, etc.

Response 1: Thanks for your kindly suggestions. We have rectified the error after through proof reading and the year and authors name have been added. We have also taken the MDPI English editing services and hoping that the typing mistakes will be eliminated.

Point 2: You need to reference and explain equations provided in text.

Response 2: Thanks for your kindly suggestions. Explanation of the equation can be found in the line from 280 to 289 and reference of the equation can now be found in the line number 231.

Point 3: In Equation 6, consider using summations.

Response 3: Thank you so much for your kindly suggestions. We used the equation in the same manner as it has been used in other research papers. We feel more comfortable to publish this in the same way as it has been done before in other papers.

Point 4: The discussions around food security indicators are not clear. By food security, do you mean the welfare of farmers? It needs a lot more clarification since food security may have many different meanings in the literature.

Response 4: we have discussed have provided more detail around food security with more references hoping it will serve the purpose.

Point 5: In Fig. 1, are the share supposed to sum up?

Response 5: The figure 1 is actually a conceptual framework and suppose to communicate the sequence and relation among variables.

Point 6: In general, the results are not well explained in a way supporting your discussions. For example, in Table 2, the results for Farm owner are significant. Does it mean farm owners tend to use risk management tools more?

Response 6: Thanks for your kindly suggestions. We worked on your suggestions to make the discussion more visible and concrete. In response to your suggestions we have also added a new paragraph at the line number 391 to 411.

Point 7: You need to explain the coefficients more quantitively rather than just describe them in discussions.

Response 7: Thanks for your kindly suggestions. The coefficients have been described quantitatively.

Point 8: You mentioned in many places the results are in line with previous study. Then, what is the core contribution of this study?

Response 8: the contribution lies in the novelty and extension of literature available presently regarding these variables in the contest of developing countries especially Pakistan. The study also provide policy implication framework with respect to selected combination risk management strategies. The contribution also well explained the conclusion and implication section.

Point 8: You mentioned sharecropper a couple times in the paper. Was it in your analysis as well?

Response 8: we just refer the sharecrop agreement where the tenant does not pay the rent but give half of the agriculture production to the landlord. Hence in some cases the term sharecropper can be interchangeably used with the tenant. But in our study tenant was our center of attention not the sharecropper.

Reviewer 2 Report

Introduction

Line 56

Multiple studies have highlighted the importance of the ex-ante risk mitigating strategy and crop 56 insurance seems a perfect example of it[16]…

Data and methods

Mention multiple studies but only cities ones.

Line 122

Muzaffargarh and Dera ghazi khan Figure 2 based on vulnerability and food insecurity….

What means vulnerability and food insecurity? How is measure it?

Line 136

At the third stage, we selected multiple villages on the base of their catastrophic nature and flooding history

The same of previous question. In this case, explain catastrophic natura and flooding history.

How many time spend in doing the survey?

Please, explain the survey, question, section…

Dependet variable is crop insurance: explain the advantages and disadvantages

Line 214 Multiple studies have indicated the crucial role farm size in the uptake of risk management approaches. Please mention the studies.

When ask about the risk perception, what exactly means the question? What type of risk refer?

Line 242-243 Based on multiple cut points, the households can be categorized into the following groups: (<21.5), borderline (21.5–35) and acceptable (>35).

One group is not mention.

Is the sample representative of some population?

Results:

Figure 4. What represent the accumulative data?

Table 1.-How is calculate the risk management instruments? Please, describe the regression and the results.

Figure 5 in line 316 should be 6. Rename the next figures.

How is use the propensity score? Please introduce the results of the applications. It is mention in the introduction but not in the rest of the paper

Author Response

Response to Reviewer 2 Comments

General Response: Respected reviewer thank you so much for your precious time and providing us your strong feedback to make our paper more productive.

Point 1: (Line 56) Multiple studies have highlighted the importance of the ex-ante risk mitigating strategy and crop 56 insurance seems a perfect example of it[16]…

Response 1: Thanks for your kindly suggestions. We have rectified the error after through proof reading and citation have been updated.

Point 2: You need to reference and explain equations provided in text.

Response 2: Thanks for your kindly suggestions. The explanation of the equation can be found in the in the lines of 270 to 289. Whereas the reference of the equation can be found in the line number 331.

Point 3: Data and methods

Mention multiple studies but only cities ones.

Line 122

Muzaffargarh and Dera ghazi khan Figure 2 based on vulnerability and food insecurity….

What means vulnerability and food insecurity? How is measure it?

Line 136

At the third stage, we selected multiple villages on the base of their catastrophic nature and flooding history

The same of previous question. In this case, explain catastrophic natura and flooding history.

How many time spend in doing the survey?

Please, explain the survey, question, section…

Dependet variable is crop insurance: explain the advantages and disadvantages

Line 214 Multiple studies have indicated the crucial role farm size in the uptake of risk management approaches. Please mention the studies.

When ask about the risk perception, what exactly means the question? What type of risk refer?

Line 242-243 Based on multiple cut points, the households can be categorized into the following groups: (<21.5), borderline (21.5–35) and acceptable (>35).

One group is not mention.

Is the sample representative of some population?

Response 3: Thank you so much for your kindly suggestions. The sample is well representative of the population of the disaster prone areas. We have use multistage random sampling technique which is in accordance of previous studies with similar nature. Furthermore the all of your questions regarding flooding, vulnerability, data collection time, crop insurance and regarding other mistakes have properly addressed.

Point 4:  Results:

Figure 4. What represent the accumulative data?

Table 1.-How is calculate the risk management instruments? Please, describe the regression and the results.

Figure 5 in line 316 should be 6. Rename the next figures.

How is use the propensity score? Please introduce the results of the applications. It is mention in the introduction but not in the rest of the paper

Response 4: Thanks for your kindly suggestions. Accumulative data is represented through percentage of associations among land tenancy types and risk management strategies. Moreover, the results on risk management instruments in table one are calculated through correlation coefficients. Furthermore, the error regarding figure number has been removed. We mentioned the propensity scores in the context of another study which has been cited the introduction. Hence the propensity scores were not the part of our analysis.

Reviewer 3 Report

I want to thank the editors of the International Journal of Environmental Research and Public Health for the opportunity to review the manuscript, “Mitigating Catastrophic Risks and Food Security Threats: Effects of Land Ownership in Southern Punjab” for their publication. I found the work an interesting and relevant work looking at what land-use practitioners in southern Punjab of Pakistan thought about and used for mitigation strategies to avoid periods of farm household hunger potential and reliance on single-stream agricultural revenue sources. I think eventually this manuscript should be published. However, the current version of the manuscript needs substantial work before it reaches the point of potential publication.

One of the largest issues is a major need for the paper to have a through formal English editorial review before its revised submission. The current version suffers from both probably unfamiliarity of writing in formal English (e.g. present and past tense sentence construction, various punctuation issues, spacing, strange capitalization of words, where to use “which” versus “that”, where to use “because of” instead of “due to”, awkward word choices, etc.) but also seemingly carelessness in re-checking the work before submission (e.g. seemingly incomplete references such as dates when proceedings were held, using “verities” multiple times for “varieties” although “varieties” is also used, etc.). The overall writing leaves the reader distracted from the main goal of getting the author(s) research across in an understandable and pleasant read.

Other issues include:

The 2.3.1.1 sub-section of “dependent variables” seems to be in an odd place after the previous section that contained model equations. I believe the information found in 2.3.1.1 is necessary but perhaps should be towards the end of the introductory materials.

The author(s) is using a sampling strategy with the farmer interviews, correct? If so, why isn’t there any uncertainty measures, such as standard error bars, on the percentage graphs of Figures 4, 5, 5 again, and 6?

For Table 2, page 9, aren’t most of the SEs for the “Improved Varieties” larger than the estimates? What does that say for the sampling on that variable? Can the author(s) really claim the ability to determine significance for various aspects of that variable with statistical uncertainty higher than the estimates?

The reference list appears to be incomplete or inconsistency done, such as how does a person access say some of the UN reports (do they have website links?), the dates when conferences or meetings were held that are the basis for proceedings presentations or reports, titles of journal articles that have just the first word capitalized whereas others have all the words in the article title capitalized. There are also odd spacing within multiple references. The whole list just needs better proof reading for consistency and completeness.

Other, more minor comments include:

Line 30- “development of agriculture”…? Economies solely or heavily dependent on agriculture tend to be some of the poorest in the world so is it more than “development” here? How about “improvement” or “modernization” or another similar word instead?

Line 35- “warned Pakistani officials” instead of “warned Pakistan”. The country is an inanimate conceptual object, it can’t do anything itself, only people within the country can act.

Line 36- “Floods” for the manuscript needs to be defined. I presume the author(s) is talking of riverine flooding, but this is not clear. Does coastal or closed-basin lake flooding also impact farming in Pakistan?

Line 42- Does “risk” need to be used twice in this sentence and then lead the next sentence? Try to reduce redundancy in the writing.

Line 44- The goal is to enhance food “security”, not “insecurity”, correct?

Line 87- You may want to define “owner-cum-tenants” in a greater context. In the United States, we have farmers that own some of their own land and rent other people’s land to operate a larger farm business, but your term is totally foreign in usage here and probably other places as well.

Line 93-94- Why is this so…?

Lines 97-98- A single sentence paragraph?

Where is the discussion and call out for Figure 1 in the text…?

Line 124- “mountain” range

Line 126- You have mentioned “natural disasters” and “catastrophes” several times, but what else besides “flooding” (undefined what kind of floods) and “heat waves” have been mentioned. What other “natural disasters” are there in the study region?

Lines 134-135. Redundant, you’ve already told the readers this previously.

Line 135- What are “tehsils”?

Line 201- What does, “…eighth standard=2, matric=3” mean?

Line 204- the term “extension” should probably be defined

Line 235- Is this acronym commonly used in the literature or is it self-defined?

Line 256- “primarily passed”…?

Line 258- try to avoid hyperbolic words, in this case “vast”

Line 261- doesn’t the English measurement “acre” need to be converted to “hectares”?

Author Response

Response to Reviewer 3 Comments

General response

Thank you so much for your precious time and highlighting the important points. Apart from your key concerns you also highlighted some minor points. We have addressed both of your major and minor points to the best of our abilities. With regard to the major points you can find our detailed answers one by one. Whereas the minor points have been addressed in the manuscript.

Point 1: One of the largest issues is a major need for the paper to have a through formal English editorial review before its revised submission. The current version suffers from both probably unfamiliarity of writing in formal English (e.g. present and past tense sentence construction, various punctuation issues, spacing, strange capitalization of words, where to use “which” versus “that”, where to use “because of” instead of “due to”, awkward word choices, etc.) but also seemingly carelessness in re-checking the work before submission (e.g. seemingly incomplete references such as dates when proceedings were held, using “verities” multiple times for “varieties” although “varieties” is also used, etc.). The overall writing leaves the reader distracted from the main goal of getting the author(s) research across in an understandable and pleasant read.

Response 1: Thanks for your kindly suggestions. To resolve the English related issues we have opted for the MDPI English editorial service. We hope that the updated manuscript is up to your English standards. 

Point 2: The 2.3.1.1 sub-section of “dependent variables” seems to be in an odd place after the previous section that contained model equations. I believe the information found in 2.3.1.1 is necessary but perhaps should be towards the end of the introductory materials.

Response 2: In response to your concern we have changed the sequence and now one can found the description of the variable at the end of introduction.

Point 3: The author(s) is using a sampling strategy with the farmer interviews, correct? If so, why isn’t there any uncertainty measures, such as standard error bars, on the percentage graphs of Figures 4, 5, 5 again, and 6?

Response 3: Regarding the pointer of the learned for using uncertainty measures. We used “stacked Bars” comparing the three factors in each category – providing a comparative picture of the risk perception and risk management amongst three categories of land ownership: owner, owner-cum-tenants, and tenants. Hereby, using such bars – representing figure 4 and 5 – we cannot indicate any uncertainty measure like standard error bars as such uncertainty measures can only be used for single factor(s) representing individual characteristics. 

Likewise, regarding Figures 6 and 7, these Figures are based on Chi-square test results given in Table 3. Here, we compared the similar categorical characteristics for food security status on food consumption score, Figure 5, and household hunger scale Figure 6 – representing the comparative scenario on adoption and non-adoption of the given characteristics. Thus, using such a categorical Illustration, one cannot use uncertainty measures.  

Point 4: For Table 2, page 9, aren’t most of the SEs for the “Improved Varieties” larger than the estimates? What does that say for the sampling on that variable? Can the author(s) really claim the ability to determine significance for various aspects of that variable with statistical uncertainty higher than the estimates?

Response 4: Corrected. We are thankful to the learned reviewer for highlighting such a contextual error. Actually, while manually reporting the results in Table, one of authors committed some mistakes, therefore, sometimes reported coefficients as standard error – henceforth corrected. Also, we have critically reviewed the actual results reported throughout the study and double-checked with data to overcome any further inconsistency, typos or loophole in reporting the study results. 

Point 5: The reference list appears to be incomplete or inconsistency done, such as how does a person access say some of the UN reports (do they have website links?), the dates when conferences or meetings were held that are the basis for proceedings presentations or reports, titles of journal articles that have just the first word capitalized whereas others have all the words in the article title capitalized. There are also odd spacing within multiple references. The whole list just needs better proof reading for consistency and completeness.

Response 5: thank you so much for your keen review and highlighting the referencing mistakes. While addressing your concerns we have revisited the referencing section and hoping that your concerns have been addressed.

Round 2

Reviewer 1 Report

Thank you for the revisions and improvements made. All my comments were addressed.

Author Response

Response to the general question: Respected reviewer thank you so much for your precious time and providing us your strong feedback to make our paper more productive. we strongly appreciate you for investing time to make comprehensive suggestions and accepting our response accordingly.

Reviewer 2 Report

I consider that in the new version of the paper the authors have made an important effort to improve it.

They have clarified the main question raised in the review satisfactorily and in my opinion I think the paper is in a position to be published in the review.

Author Response

(The authors gave the same response as above.)

Reviewer 3 Report

I think the manuscript is largely improved. Most of my issues are still editorial in nature (see the complete list). I do wonder if the "Discussion" section is too long. The "Discussion" shouldn't be a second literature review. It should be how your research improves the overall knowledge base of the subject and the strengths and contrasts of where it has different views of previous work.

Line 30- “modernization” or “improvement” are better word choices than “development” of agriculture

Line 35- have “warned the Pakistani government” or “Pakistani officials”, can’t just warn the country, it’s an inanimate thing.

Line 47- I think we know you’re talking about “risk management” already (used 4 times in 2 sentences), I would delete “risk management” in this line and use “these approaches” instead.

Line 76- make “farmer” plural

Line 79- drop the last “are”. It is not needed.

Line 99- “losses” instead of “loses” (?)

Where is the call-out to Figure 1 in the text…?

Lines 124-125- “scheme” tends to have a pejorative use in English, at least U.S. English. I would use “program” and “option” instead in this sentence. You also don’t need the second “are”. So maybe in line 124, “…crop-insurance programs available to Pakistani farmers…”, And finishing that sentence in line 125, “…crop loan insurance option.”

Line 128- “A farmer” instead of “The farmers”

Line 130- “largest” instead of “biggest”

Line 134-135- How does crop insurance “enhance yields”? I would drop this part of the sentence.

Line 141- I would use the term “helps mitigate” instead of “protects from”. There are some climatic shocks that improved varieties won’t protect from, even if used.  

Line 180- words are missing to make this sentence complete, “Risk perception is awareness of climatic risks such as…” (?)

Line 183- can be shortened to “…where 1 represented the lowest perception and 5 the highest perception.”

Is Figure 3 now supposed to be Figure 2? The old Figure 2 has been moved further in the text so the figure numbering needs to be updated.

Line 200- what does “cut” mean?

Lines 204-205- These sentences can be improved with some editing. I would delete “at all” in line 204 and replace “no resources” with “inadequate resources”. I would delete “at night” in line 205. Not needed, almost all people sleep at night.

Line 215- “because of” instead of “due to” (?)

Line 218- I would add “mountain” in front of “range” here

Line 221- “…and house some…” comes across awkward. How about, “…and is home to some of…” (?)

Line 227- I don’t think “still” is needed. I would add “other” in front of “industries” in this sentence.

Line 233- I would modify to be “…6 months of recent survey…”

Line 234- “tehsils” is still undefined

Line 236- what kind of “infrastructure”…? Maybe you can just put “other” in front of “infrastructure” you have there now

Lines 242-247- You use the term “risk management” 5 times in 5 lines. It seems a bit of “over-kill” for this term. Try to rework the text a bit so people know you’re talking about “risk management” but you don’t have it say it all the time.

Line 268- The term “vast” seems to be “too much” for this situation. I would use “substantial experience in farming…”

Lines 274-275- are “pests” plant diseases”? “Pests” is usually used for insects and other animal lifeforms. I would change to “…pests and plant diseases”.

Line 286- I don’t think the “are” after “owners” is needed

Line 292- Instead of repeating all the risks in the sentence above, you could say, “…were concerned about the same risks, respectively.”

Line 298- I don’t think you need to use “risk management” again in this sentence, just say, “…of other available choices” or some other adjective of “choices”

Line 317- “improved varieties”, not “improved verities”

Line 323- “This shows”, not “Which shows”

Line 324- “improved varieties”, not “improved verities”

Line 326- “improved varieties”, not “improved verities”; “unit” made plural (?)

Line 361- I would delete “the” in front of “farm owners”, it’s not needed

Line 363- You use a form of “similar” twice within 5 words. I would start this sentence with “Likewise” instead

Line 364- You have a modifier in front of “wealth” but none of the other nouns in this sentence. I would change by adding “better household welfare” (drop “and” in front of “household”), and “increased control”

Line 365-366- “Because of” instead of “Due to”

Line 367- Again “scheme” is often taken as a pejorative in English, at least U.S. English, so I would change to “crop-loan insurance plans than other insurance options.”

Line 370- “tenants seem reluctant”

Line 373- “because of” instead of “due to”

Line 377- “…greater farming experience” instead of using “increases” twice in the same sentence

Line 389- “that shows” instead of “.., which shows”, also “the” not needed in front of “farmers”

Line 389- “huge” is probably “too grand” of a word choice, how about just “large” instead?

Lines 403-404- instead of using “crop insurance” again in the same sentence, just end the sentence with, “…encourages farmers to adopt it.”

Line 407- I would use “increasing” instead of “growing” in front of “age”

Line 413-414- You use a form of “improve” 3 times in this single sentence, think of other word choices!!

Line 435-436- You probably don’t have to repeat the whole previous sentence to say that Ullah et al. 2019 found it as well. You can truncate this sentence by saying, “Similarly, Ullah et al. 2019 found the same relationship.”

Line 443- drop “the” in front of “farmers”

Line 455- “Researchers suggest” instead of past tense

Line 467- Because you have already said “risk-management strategies” a few words before, no need to state it again. I would delete the second “risk-management” and just say “such strategies”

Line 483- make “land-right” plural (?)

Line 487- Again, I wouldn’t use the word “huge”. “Large” will suffice.

Line 488- “agro ecological” probably needs to be hyphenated

The call out to Figure 8 in the text?

Lines 517-518- How does a reader access this reference? By the way, a “unique” way to reference the United Nations…

Lines 558-559- Reference appears to be incomplete. How does a reader access it?

Lines 560-561 same as above

Lines 576-577- Reference appears to be incomplete. Was this a conference, if so when was it held? How does a reader access it?

Lines 578-579- Reference appears to be incomplete. Was this a published thesis or dissertation…?

Line 588- same as above

Lines 589-591- Reference appears to be incomplete. If this is a chapter in a book, what are the page numbers? If it’s a book, who published it?

Lines 597-598- Reference appears to be incomplete. Was this a published thesis or dissertation…?

Lines 602-603- same as above

Lines 612-614- How does a reader access this publication?

Lines 624-625 same as above

Lines 642-644- Dates this conference was held? How does a reader access this publication?

Lines 648-649- same as above

How does the journal handle the titles of journal articles, should they have capitalization only for the first word of the title (and perhaps formal names) and then lower cases for the rest of the words in the title? Check with journal editor.

Author Response

Response to Reviewer Minor Comments

Response to the general question: Respected reviewer thank you so much for your precious time and providing us your strong feedback to make our paper more productive. On the part of your suggestions, we have tried to incorporate all your suggestions. We have rechecked the reference section to add all the suggested information. The reference regarding figure number 1 and 8 can now be found at the line number 119 and 534. Furthermore, the clarification regarding tehsil can found in line number 259. Hoping we have addresses all of your key concerns.
